# Recent Advances in Directed Yeast Genome Evolution

**DOI:** 10.3390/jof8060635

**Published:** 2022-06-15

**Authors:** Zhen Yao, Qinhong Wang, Zongjie Dai

**Affiliations:** 1Key Laboratory of Systems Microbial Biotechnology, Tianjin Institute of Industrial Biotechnology, Chinese Academy of Sciences, Tianjin 300308, China; yaozh@tib.cas.cn; 2National Center of Technology Innovation for Synthetic Biology, Tianjin 300308, China

**Keywords:** *Saccharomyces cerevisiae*, evolutionary engineering, synthetic biology, CRISPR/Cas9, SCRaMbLE

## Abstract

*Saccharomyces cerevisiae*, as a Generally Recognized as Safe (GRAS) fungus, has become one of the most widely used chassis cells for industrial applications and basic research. However, owing to its complex genetic background and intertwined metabolic networks, there are still many obstacles that need to be overcome in order to improve desired traits and to successfully link genotypes to phenotypes. In this context, genome editing and evolutionary technology have rapidly progressed over the last few decades to facilitate the rapid generation of tailor-made properties as well as for the precise determination of relevant gene targets that regulate physiological functions, including stress resistance, metabolic-pathway optimization and organismal adaptation. Directed genome evolution has emerged as a versatile tool to enable researchers to access desired traits and to study increasingly complicated phenomena. Here, the development of directed genome evolutions in *S. cerevisiae* is reviewed, with a focus on different techniques driving evolutionary engineering.

## 1. Introduction

*Saccharomyces cerevisiae* has drawn increased attention due to a number of advantages, such as high robustness to industrial conditions, feasible molecular manipulation and natural resistance to phage invasion [1,2,3,4]. Many successful endeavors, including construction and optimization of heterogenous pathways [5,6], as well as modification of chassis cells [7,8], have been devoted to obtain a superior chemical producer and to study physiological phenomena. As a result, *S. cerevisiae* has become one of the most promising model chassis cells for both industrial applications and basic research. Among these processes, it is unavoidable to introduce genetic variants such as insertion, replacement and/or deletions to cause perturbation in phenotype.

Straightforward methods for increasing the production of chemicals include the overexpression of genes from biosynthetic pathways and the downregulation of genes from competitive branches. However, identifying rate-limiting steps in biosynthetic pathways requires iterative and time-consuming operations. Moreover, cellular regulations can be quite unpredictive as, for example, genes from noncanonical pathways that are responsible for specific functions are identified differently due to the limited knowledge about the complexity of genetic backgrounds and intertwined metabolic networks. Similarly, it can be challenging to predict genes involved in complicated phenotypes such as cell growth and tolerance to environmental stresses. Therefore, instead of rational manipulation of predicted loci to obtain specific traits, genome evolution engineering through serial propagation [9] and synthetic methods [10] can be applied to facilitate this progress by randomly introducing mutations into the genome. In this case, compared with rational design, knowledge of relevant background is not required. Through genome evolutionary engineering, mutations are embedded to endow strain-specific traits.

The essence of genome evolutionary engineering is about randomly introducing various mutations or structure arrangements into the genome, followed by screening under various selective pressures. The methodologically straightforward method adaptive laboratory evolution (ALE) is the most popular technique for improving traits of interest with aid of spontaneous mutations inherent to DNA-replicating reaction, which was well-reviewed in a previous study [9]. Alternatively, similar to the error-prone PCR, cellular DNA replication with disrupted proofreading activity has drawn much attention towards increasing the efficiency of genome mutation. Finally, with the continuous development of new and improved techniques in molecular biology, more and more refined methods are being generated to achieve genome evolution, with some examples being CRISPR/Cas9 and SCRaMbLE. In this review, the techniques behind various genome evolutionary-engineering methods are highlighted.

## 2. DNA-Replication Protein-Related Methods

During DNA replication, mismatched bases are rarely introduced into double strand due to the proofreading function of 3′ to 5′ exonuclease activity [11] as well as the recognition accuracy of DNA polymerases [12]. Furthermore, mismatch bases that escaped from DNA polymerases’ high fidelity are further subjected to cellular DNA mismatch repair (MMR) [13]. All these correction processes avoid considerable changes in genetic information through generations. However, low spontaneous mutation rate necessitates the long duration of ALE—usually months to obtain access to adaptive strains with desired traits. Since DNA replication, proofreading and MMR happen in series, interference with any of these steps would increase the mutation rate of synthetic DNA strands. For instance, researchers have increased DNA mismatching efficiency in various ways, including generating DNA polymerase mutants with deficient accuracy [14] and proofreading function [15], as well as blocking the serial execution of cellular MMR [14,15].

In *S. cerevisiae*, there are two main DNA polymerases responsible for DNA strands extension, namely Pol ε and Pol δ, which act as complexes that extend the opposite leading and lagging strands of replication forks, respectively [16,17]. Theoretically, *S. cerevisiae* that contains Pol ε or Pol δ mutants with deficient proofreading functions can equally enhance mutation efficiency due to the same importance of each DNA strand. However, in practice, Pol δ mutants with deficient proofreading function show higher mutation rates than Pol ε ones because wild-type Pol δ can somehow repair mismatched bases introduced by Pol ε mutant, with similar capabilities not displayed by wild-type Pol ε [18]. As such, methods of engineering DNA polymerase to increase mutation rates focus on Pol δ.

The subunit Pol 3 of Pol δ contains catalytic and 3′ to 5′ exonucleolytic proofreading domains. Amino-acid residue Leu612 of this subunit plays a vital role in keeping the accuracy of DNA replication by packing against Tyr613, which shapes the binding pocket and directly contacts incoming dNTPs [12]. Single mutations of Leu612Met or Leu612Gly increased L-canavanine-resistance forward-mutation efficiency by 24 and 1100 times, respectively [14]. On the other hand, Pol 3 mutants with defective proofreading functions increased the possibility of maintaining mismatched bases within the genome, as inheritance of the offspring. Such a defective proofreading Pol 3 mutant *pol3-01* was created by changing the predicted 3′–5′ exonuclease active motif Phe-Asp-Ile-Glu-Cys to Phe-Ala-Ile-Ala-Cys [15]. Mutant *pol3-01* increased the reversion-mutation efficiency of His^−^ to His^+^ by 240 times and the forward-mutation efficiency of Ura^+^ to Ura^−^ by 130 times. By replacing *pol3* with mutant *pol3-01*, a thermotolerant *S. cerevisiae*, capable of proliferating at 40 °C, was successfully isolated [19]. Combinations of accuracy and proofreading deficient residues to generate Pol δ variants with defective proofreading function and accuracy further increased the mutation efficiency of genome [14].

Even after escaping from the proofreading functions of DNA polymerases, the error bases will be subjected to cellular MMR, which is responsible for recognizing and correcting mismatched bases so as to maintain genome stability [20]. The fact that deficient MMR has been implicated in multiple cancers [21,22,23] suggests its participation in increased mutation rate. The protein encoded by *MSH2* binds mismatched DNA and triggers the following MMR [24], while the ATP-binding protein encoded by *PMS1* is required in meiosis, which has also been implied in involvement in MMR [25,26]. Thus, generating a deficient MMR by knocking out MSH2 resulted in a 270-fold increase in L-canavanine-resistance forward-mutation efficiency [14]. Similarly, knocking out *PMS1* resulted in a 240-fold increase in reversion-mutation rate of His^−^ to His^+^ [15]. Furthermore, combinations of Pol 3 variants and deficient MMR showed multiplicative increases in mutation efficiency, indicating synergistic effect between mismatched base introduction and deficient MMR. However, this mutation efficiency is catastrophic for haploids, so combinations were commonly harnessed in diploids.

To explore methods that would be more suitable for haploids and extend the available toolkit for genome evolution, fusion proteins consisting of DNA-replication-related proteins and cytidine deaminase were developed to conduct directed genome evolution in yeast. Cytidine deaminase has been commonly used to transform cytosine to guanine in base editor, with nCas9 as the part of the fusion protein [27,28]. Instead of fusing with nCas9, cytidine deaminase can also be fused with varied proteins related in DNA replication including replication factor A, DNA primase, DNA helicase A or topoisomerase I to establish a random base editing (rBE) [29]. These proteins carry cytidine deaminase to close to DNA where single-stranded DNA (ssDNA) temporarily exists during DNA replication, providing the prerequisite to transform C to G by cytidine deaminase (Figure 1a). By performing rBE in *S. cerevisiae*, β-carotene production was increased to 2.4 times higher compared with the control strain.

Helicases are enzymes that unwind DNA by breaking the hydrogen bonds between bases to form ssDNA [30]. A novel fusion device of yeast helicase Mcm2-7 subunit MCM5 and cytidine deaminase was built in *S. cerevisiae* to increase mutation rates [31]. After 8 rounds of culturing *S. cerevisiae* expressing MCM5-cytidine deaminase, specific production of β-carotene was increased by 75% compared with the parental strain. In fact, the expression of cytidine deaminase only also increased mutation rate, although not as high as that of the fusion protein, implying the vital role of ssDNA in base transformations by cytidine deaminase.

## 3. CRISPR/Cas9 Driving Genome Evolutionary Engineering

Repression, activation and deletion of genes cause perturbations in transcriptional levels. These perturbations are inevitable for obtaining access to evolved strains and implementing genetic interrogations to understand genotype–phenotype relationships. This necessitates methods that are capable of precisely targeting specific genes. In this context, the CRISPR (clustered regularly interspaced short palindromic repeats) system leads DNA endonuclease Cas9 to cleave targets through the assistance of small guide RNA (sgRNA) with homology to the specific loci [32,33], and this approach provides a versatile platform for evolutionary engineering. Recent advances in phenotype screening by CRISPR/Cas system focused on combining up- and downregulations as well as sgRNA design, which are summarized in Table 1.

### 3.1. Combiantions of Up- and Downregulations

Catalytically dead Cas9 (dCas9) without cleaving activity sterically blocks the binding and/or elongation abilities of RNA polymerase, hence leading to downregulated gene-expression levels, referred to as CRISPR interference (CRIPSRi) [33]. This proof of concept has been demonstrated in *S. cerevisiae* via the fluorescence intensity of green fluorescent protein (GFP) controlled by *TEF1* promoter. Expression of dCas9 along with sgRNA targeting *TEF1* promoter resulted in 18-fold repression in fluorescence intensity [34]. In addition, fusing dCas9 to the transcriptional repressor, Mxi1, further aggravating the fluorescence repression to 53-fold.

Compared with alternative expressions of transcriptional repressor in CRISPRi, CRISPR aviation (CRISPRa) requires strict fusion with an activation domain to enhance transcriptional levels [35]. The activator VP64 was fused to dCas9 to increase transcriptional levels by targeting dCas9-VP64 to natural and artificial promoters. A hybrid VP64-p65-Rta tripartite activator (VPR) with stronger intensity was fused to dCas9 in *S. cerevisiae*, leading to a broad range of activation from 5- to 300-fold than the based activator VP64 [36].

Moreover, completely deleting the open reading frame (ORF) of the target gene was used to determine its effect on specific phenotype. The method CHAnGE was built by randomly knocking out ORF using fully catalytic Cas9 and synthetic arrays consisting of gRNA and knocking out recombination donors. The gRNA and donor were both embedded into plasmid to avoid loss during culturing. Through two rounds of transformation of the synthetic DNA, optical density (OD) at 600 nm in 10 mM furfural was increased by 8.1-fold [37].

The above-mentioned methods have been successfully implemented in up- and downregulations separately. However, sophisticated phenotypes usually involve multiplex genes in both upregulated and downregulated ways, and hence they require methods that are capable of simultaneously realizing overexpression, repression and deletion. The method STEPS, based on two fusion proteins consisting of dCas9-Mxi1 and dCas9-VPR, realizes the graded expression of targeted genes. STEPS has been successfully used to define the limiting steps of glycerol fermentation, 3-dehydroshikimate production and xylose catabolism [38]. A more facile method is that of MAGIC, which consists of a combination of three functions to control the expression level of genes throughout the whole genome of *S. cerevisiae* [39]. Three different protein structures, dLbCas12a-VP, dSpCas9-RD1152 and SaCas9 with functions of activation, repression and deletion use three specific sgRNA libraries to guide the protein structures to the genomic loci. By performing MAGIC in *S. cerevisiae*, genes related to complicated phenotypes such as furfural tolerance and protein-surface displaying were identified.

### 3.2. SgRNA Design

The ability to target specific loci with the CRISPR system through sgRNA makes it appealing for designing and constructing sgRNA libraries to satisfy different requirements, including targeting scopes and graded transcriptional levels.

At first, the sgRNA scope is important for phenotype screening because it directly controls genes subjected to expression perturbations, thus impacting the possibility to obtain access to desired phenotypes. Simple phenotypes, such as carbon catabolism and chemicals biosynthesis, are easy to focus on for genes in relevant pathways [38]. However, when it comes to more complicated phenotypes, such as cell growth and resistance to environmental stresses, more genes such as transcription factors and protein kinases are selected as targets for perturbation, because these proteins control global genes spanning from synthetic pathways to cellular signaling [40]. While it is impossible to precisely predict genes involved in some phenotypes due to a limited knowledge of cellular networks, whole-genome coverage by an sgRNA library bypassed this requirement by targeting all genes within the genome. Designed oligonucleotides can be synthesized on a chip to form a pool consisting of sgRNA that cover most—if not all—genes, and this method was shown to reach more than 100-fold coverage to ensure that every gene was efficiently targeted [37]. In the trifunctional CRISPR system MAGIC, sgRNA libraries with genomic coverage vary from each other in length, and dCas12 was used to accomplish transcription activation. Altogether, these features lead to an orthogonal and multipipeline system [39].

With extension of the sgRNA scope to a genomic scale, rapidly and preciously identifying genetic variants responsible for the observed phenotypes is usually time-consuming and labor-intensive. The method MAGESTIC uses array-synthesized guide–donor oligos to target multiplex genes at the genome scale, with genome-integrated barcodes helping to achieve precise identification of one-to-one regions in *S. cerevisiae* [41]. The genome-integrated barcodes prevent the loss of plasmid barcodes and rapidly phenotype robustly. MAGESTIC will be widely used in revealing relationships between genotypes and phenotypes.

On the other hand, it is more difficult to predict the extent to which transcription levels need to be optimized to develop various phenotypes of interest. It has also been widely indicated that the location of dCas9 of dCas9-Mxi1 on different regions of promoter appeared different repressive effects [38]. Even for CRISPRa architecture dCas9-VP64, binding of the fusion protein to sequences spanning the TATA box and the Kozak sequence was shown to repress GFP expression [35]. Optimizing the transcription by adopting graded levels was also achieved by designing sgRNA targeting various locations in promoters. Few studies use CRISPRa alone to screen phenotypes due to the inability to predict the activated location of gRNA, and improper location leads to suppression [42].

**Table 1 jof-08-00635-t001:** Genome evolutionary-engineering methods by CRISPR/Cas9.

Name	Description	gRNA	Applications	Reference
CRISPRi	dCas12a-Mxi1	Targeting heterologous β-carotene biosynthesis pathway genes crtE, crtYB, crtI	β-carotene production	[43]
	dCas9	Targeting *PFK1* and *PYK1*	N-acetylglucosamine production	[44]
	dCas9	Targeting seven genes in branch pathways of β-amyrin production	β-amyrin production	[45]
	dCas9-Mxi1	Targeting over 98% of essential and respiratory growth-essential genes	Acetic-acid Tolerance	[46]
	dCas9-Mxi1	Targeting transcription start site (TSS) in genome scale	Mining of haploinsufficient genes and identification of adenine and arginine biosynthesis genes	[47]
	dCas9-Mxi1	Targeting 161 transcriptional factors and 129 protein kinase	Growth in lignocellulose hydrolysate	[40]
CRISPRa	dCas9-VP64	Targeting 52 genes	Thermotolerance	[48]
CHAnGE	Global deletion of genes from genome by homologous recombination via gRNA and donor synthesized on chip	Synthesized on chip in ~100% gene coverage	Acetic acid and furfural tolerance	[37]
STEPS	Combination of dCas9-Mix1 and dCas9-VPR to simultaneously up- and downregulate transcriptional levels, respectively	Graded targeting of genes involved in glycerol production, PPP genes for 3-DHS production and xylose catabolism	Glycerol and 3-DHS production, xylose catabolism	[38]
CRSPRi/a	Cas9-VPR	Four genes *HMG1*, *ERG9*, *DPP1*, and *UPC2*	α-santalene production	[42]
CRSPRi/a	dCas9 or dCas9-Mxi1dCas9-VP64 or dCas9-VPR	Targeting four genes *HRK1*, *SSK2*, *ISC1* and *BDH2*	Tolerance towards lignocellulosic hydrolysate.	[49]
MAGIC	Combination of dLbCas12a-VP, dSpCas9-RD1152 and SaCas9 to simultaneously upregulate, downregulate and delete genes, respectively	Synthesized on chip in ~100% gene coverage	GAL7 and HED1 expression levels	[39]

## 4. SCRaMbLE Driving Yeast Chromosomal Rearrangement

Efforts to enable changes in DNA and at transcriptional levels in specific loci were shown to be a quick means of evolving microorganisms. However, chromosome rearrangements, causing genome variations on a broader scale, cannot be achieved by the aforementioned methods. The site-specific Cre recombinase causes changes, including deletions, insertions, duplications, inversions and/or translocations at chromosome level by generating recombination events between loxP sites. Symmetrical loxP sites were designed and inserted into synthetic chromosomes, and *S. cerevisiae* transformed with synthetic chromosomes was subjected to recombination events to form chromosomal rearrangements, referred to as SCRaMbLE (synthetic chromosome rearrangement and modification by loxP-mediated evolution) [50]. Different from other evolutionary engineering methods, SCRaMbLE forms chromosome rearrangements, resulting in large-scale variations of genomic changes. Research about developments of SCRaMbLE highlighted the genomic diversity and stability.

### 4.1. Diversity

Compared with canonical loxP sites, symmetrical 34-bp loxPsym sites allow recombination events in either direction to cause deletion and inversion equally. To increase the structural diversity of SCRaMbLE events, numerous loxPsym sites were embedded into a series of chromosomes, and synthetic chromosomes were subjected to Cre recombination [50,51]. Heterozygous diploid SCRaMbLE system extended the limit of recombination templates to larger scales, benefiting for reliving the lethality and increasing the genomic stability [52]. Collectively, number and variety of synthetic chromosomes all play vital roles in genomic diversity.

In early studies involving SCRaMbLE, only synthetic or semisynthetic chromosome arms were transformed into yeast, resulting in limited regions being subjected to Cre recombinase function [50,51]. In subsequent ones, whole chromosomes were then replaced by synthetic ones [53,54,55]. Researchers have combined more than one synthetic chromosome in a single strain to perform scrambling and generate diverse genetic rearrangements [56,57,58]. Recently, ref. [59] conducted SCRaMbLE evolution in a strain containing six synthetic yeast chromosomes and detected 260,000 rearrangement events [59].

The topologically structural variants influence the steric hindrance between the two loxP sites. A ring chromosome used in SCRaMbLE was shown to form different genomic rearrangements, leading to a larger combination of genetic variants that enhanced phenotypic diversities [60]. Compared to the linear chromosome, the ring-chromosome background generated relatively complex rearrangements due to decreased steric hindrance. The number of synthetic chromosomes determined the scrambling range.

### 4.2. Controlling the Expression of Cre

Cre recombinase was initially fused to the murine estrogen-binding domain (EBD) and nucleus localization was estradiol-inducible by binding β-estradiol to EBD [50]. However, residual activity of Cre was also observed in absence of β-estradiol, leading to genome instability. To strictly control expression of Cre, ref. [61] built a genetic ‘‘AND gate’’ switch for SCRaMbLE (Figure 1c). This “switch” was turned on only when both galactose and β-estradiol were present in the medium [61]. In addition, L-SCRaMbLE was constructed by dividually fusing the N- and C-terminals parts of recombinase Cre with the chromophore-binding photoreceptor phytochrome B (PhyB) and interacting factor PIF3 [62]. PhyB and PIF3 would interact with each other when exposed to red light in the presence of the chromophore of the PhyB photoreceptor, chromophore phycocyanobilin (PCB), thus restoring the function of the Cre (Figure 1c). In this way, SCRaMbLE became a red light-controlled recombination tool in yeast. This approach was shown to regulate the activity of Cre 179-fold. The exposure time and PCB concentration are both feasibly customized to adjust SCRaMbLE strength for screening different phenotypes of interest.

### 4.3. In Vitro SCRaMbLE

The reason for genomic instability is a continuous and leaking expression of recombinase Cre. Conducting the recombination reaction in vitro is able to completely overcome this instability. Furthermore, in vitro construction of DNA libraries provided a straightforward method to investigate genotype–phenotype relationships and to optimize biosynthetic pathways.

Ref. [63] developed two in vitro SCRaMbLE tools for the construction of DNA libraries, namely the top-down and bottom-up methods [63]. The former was originally constructed from an initial plasmid harboring the β-carotene pathway along with loxP dividing genes (Figure 1d). Purified Cre was used to SCRaMbLE the initial plasmid to construct the structurally varied plasmid library. On the other hand, bottom-up in vitro SCRaMbLE originally constructed different pathway-transcription units (TU) flanked by loxP, which were stochastically recombined with loxP site, as well as building a library consisting of different Tus (Figure 1d). The top-down library was transformed into *S. cerevisiae* with β-carotene production as a reporter, leading to a 5.1-fold increase in yield; and the bottom-up library led to a further increase in yield. Ref. [64] combined in vitro and in vivo SCRaMbLE to construct the SCRaMbLE-in method. In this case, regulatory elements were inserted into pathways of interest by in vitro recombination. The resulting library with diverse expression levels of related genes was then transformed into Sc 2.0 *S. cerevisiae*, followed by in vivo SCRaMbLE, to optimize genomes. It turned out that SCRaMbLE-in increased violacein and β-carotene production 10-fold and 2-fold, respectively.

Chromosomal engineering through SCRaMbLEing synthetic chromosomes generates genetic variants with phenotypic diversities and has been proven to be a versatile tool to improve a broad range of traits, including carbon source catabolism, synthetic pathway optimization, environmental stress resistance and product toxicity tolerance. With an increasing number of synthetic chromosomes in single haploid or diploid *S. cerevisiae*, deeper and more complex rearrangement events are being observed, leading to more complicated genomic structural variants and the increased diversity of phenotypes. In the future, it is anticipated that SCRaMbLEing a full set of synthetic chromosomes has the potential to extend the boundary of genome engineering and deepen our understanding of genome evolution. Moreover, the successful application of SCRaMbLEing system in *S. cerevisiae* paves the way for synthesis of chromosomes in other and even higher species.

## 5. Evolution in Transcriptional Levels

Mutations up- and downregulate the transcription of corresponding loci. The essence of evolutionary engineering is about achieving different types of mutations that regulate a single or multiple genes involved in different phenotypes of interest. Therefore, it is easy to come up with strategies to regulate genes at genomic scale to cause perturbations in phenotypes.

Zinc fingers were found in transcriptional regulation factors, acting as DNA-binding domains, with their function being to recognize specific DNA sequences. Ref [65] developed a method in which multiple zinc finger domains were fused with artificial transcription factors to induce transcriptional diversity in yeast [65]. In the fusion-expression library, zinc fingers with various DNA specificities were randomly linked with activation or repression transcription factor, thus forming 100,000 zinc-finger-factor fusion proteins. Yeasts transformed with the resulting library were screened for diverse phenotypes, such as ketoconazole tolerance, thermotolerance and osmotic shock resistance.

Global transcription-machinery engineering (gTME) is an approach for reprogramming gene transcription to elicit cellular phenotypes, which could be important for technological applications [66]. This method obtains access to a new type of diversity at the transcriptional levels by altering key proteins regulating the global transcriptome level. It turned out that through gTME and mutagenesis of the transcription factor Spt15p, an increased ethanol tolerance and more efficient glucose conversion to ethanol was achieved in *S. cerevisiae*.

After recognizing the significance of the transcription factor Spt15p in controlling phenotypes, researchers focus on engineering this protein. The *SPT15* gene encodes an *S. cerevisiae* TATA-binding protein that is able to globally control the transcription levels of various metabolic and regulatory genes. An *SPT15* gene mutant (S42N, S78R, S163P and I212N) was expressed in *S. cerevisiae* and it was observed that the mutant-expressing strain showed a higher glucose-consumption rate and ethanol productivity compared with the BSPT15wt strain [67]. On the other hand, different mutants of *SPT15* were obtained and investigated to reveal their effects on different tolerance capabilities. Ref. [68] leveraged Target-AID (activation-induced cytidine deaminase) base editor to enable C-to-T substitutions of *SPT15* and obtain 36 mutants with various stress tolerances [68]. By screening the expression of these *STP15* mutants, tolerance to a number of stresses including against hyperosmotic, thermal and ethanol stresses were shown, and at the same time, 1.5-fold increases in fermentation capacities were generated.

The above methods use zinc fingers or specific transcriptional factors to regulate multiple genes. However, these factors cannot obtain access to genome-scale coverage. Moreover, the determinant genes responsible for phenotype of interest were difficult to be identified due to unchanged DNA sequence and changed transcriptional levels of multiple genes. Ref. [69] reported an automated platform for multiplex genome-scale engineering in *S. cerevisiae* [69]. Standardized genetic parts encoding overexpression and knockdown mutations of 90% yeast genes are created in a single step from a full-length cDNA library. Bidirectional integrations of cassettes from cDNA simultaneously provided overexpression and knockdown modulation parts (Figure 1e). Combined with the CRISPR-Cas system, modulation parts are iteratively integrated into the repetitive sequence of *S. cerevisiae* genome, and through this method cellulase expression, isobutanol production, glycerol utilization and acetic-acid tolerance were effectively enhanced [69]. Gene targets were readily identified by NGS analysis.

## 6. Summary and Outlook

Genome evolutionary engineering is powerful tool to examine genes at the genomic scale and irrationally improve desired phenotypes. With the development of emerging technology such as the CRISPR/Cas system and the long-read sequencing method, sophisticated combinations have performed efficiently in facilitating evolutionary engineering, and the applications of genome evolutionary engineering in both laboratorial and industrial ways have been widely expended. Multiple methods have been combined to accomplish genome engineering. For instance, CRISPRa identified the specific loci from 52 genes with upregulated expression levels in thermotolerance phenotype caused by transcriptional factors [48,70]. Genome evolutionary engineering not only accelerates the process of obtaining mutants with desired traits but also provides insights into relationships between genetic variants and changes in phenotypes. Downstream high-throughput screening methods, such as microdroplet- and fluorescence-associated cell sorting, can further facilitate genome evolutionary engineering to obtain access to improved strains. Furthermore, in order to achieve a lower workload while applying an increasing number of genome engineering methods, automated platforms are being used to iteratively cause mutations, thereby accelerating the overall process.

CRISPR- and RNA-assisted in vivo directed evolution (CRAIDE) [71], in vivo continuous evolution (ICE) [72] and eukaryotic multiplex automated genome engineering (eMAGE) [73] normally accelerated the evolution process of specific proteins or pathways instead of the whole genome in yeast. However, when employing these methods to evolve transcriptional factors, it is highly possible to cause diverse phenotypes. Among these methods, eMAGE was derived from MAGE, which was originally implemented in *E. coli* [74]. In this context, procaryotic organisms derived methods, such as phage-assisted continuous evolution (PACE) [75], phage- and robotics-assisted near-continuous evolution (PRANCE) [76], and automated continuous evolution (ACE) [77] may also have the potential to be ported over to yeast and expand the toolkit for the genome evolution of yeast in the future.

## Figures and Tables

**Figure 1 jof-08-00635-f001:**
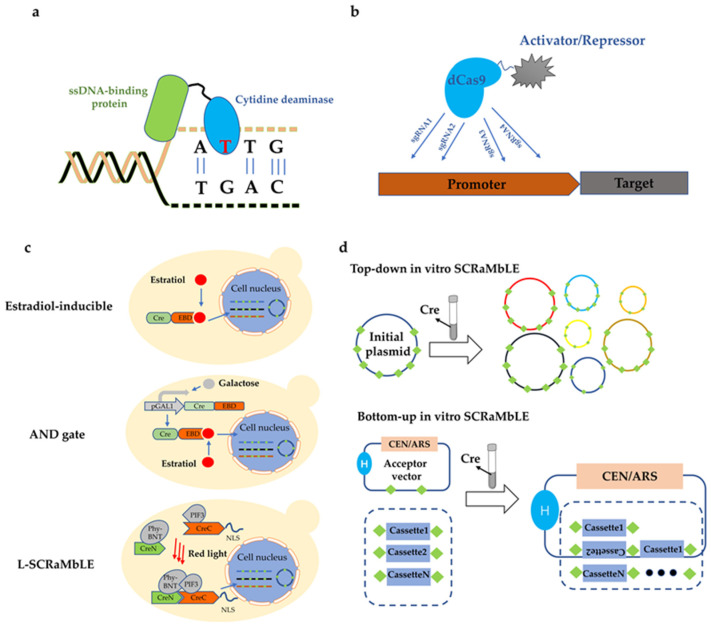
Summary of genome evolutionary engineering. (**a**) Fusion of ssDNA-binding protein and cytidine deaminase. (**b**) CIRSPRi/CRISPRa. (**c**) SCRaMbLE in vivo provides a versatile platform to generate diverse genetic variants. (**d**) SCRaMbLE in vitro is conducted via purified Cre in tube and the obtained library is transformed into *S. cerevisiae* to generate diverse phenotypes. (**e**) Directional cloning obtained from cDNA simultaneously realized up- and downregulations.

## Data Availability

Not applicable.

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
