# Peer review of "Recent Advances in Directed Yeast Genome Evolution"

_jof, 2022, doi:10.3390/jof8060635_

Round 1

Reviewer 1 Report

The review by Yao, Wang and Dai presents a good overview of the methods and contemporary approaches to directed genome evolution. While the verbiage can be sometimes overly sophisticated and it can slow down the reading flow, that doesn't take away from the very good breadth and detail of the discussed approaches such as CRISPR and SCRAMBLE. The only aspect that I find lacking is mentioning and discussing automated methods for directed evolution such as ACE, PRANCE or MAGE. While they are targeted towards bacteria more than yeast, I think those approaches are still worth at least mentioning, since they could be ported over to the model eukaryotic system too.

Reviewer 2 Report

In this manuscript, Yao, Wang, and Dai present a comprehensive review of the recent advances in genome editing and evolution technology applied to the model yeast Saccharomyces cerevisiae with the scope to obtain tailor-made yeast strains for the precise determination of relevant gene targets that regulate physiological functions relevant to important functions such as stress resistance or metabolic pathway optimization.

The review is very well written, covering important aspects and techniques used for precise evolutionary engineering, as the primary step for synthetic biology and metabolic engineering applications. In the context of Synthetic Yeast Genome Projects, the authors focus on reviewing the use of cutting-edge techniques such as CRISPR/Cas9 or SCRaMbLE applied to yeast for laboratory and industrial purposes.

In this reviewer's eyes, the manuscript can be accepted for publication. One minor issue: the last paragraph before Summary and Outlook (lines 376-380) requires a reference. 

Reviewer 3 Report

Yao, Wang, and Dai have conceived and written an accurate review on the current state of genome manipulation aimed at modulating or investigating genetic functions in the budding yeast Saccharomyces cerevisiae.

The manuscript “Recent advances in directed yeast genome evolution” is interesting and properly depicts the current tools and approaches suitable for dissecting the links between genetics and phenotype as well as for tuning specific traits.

I have only a couple of very minor comments:

L12 (abstract): what do the authors mean by “evolution technology”?

L382: I am not sure the verb “interrogate” is appropriate here.
